# Prognostic Value of *CD200R1* mRNA Expression in Head and Neck Squamous Cell Carcinoma

**DOI:** 10.3390/cancers12071777

**Published:** 2020-07-03

**Authors:** Hyun Chang, Yun-Gyoo Lee, Yoon Ho Ko, Jang Ho Cho, Jong-Kwon Choi, Keon Uk Park, Eun Joo Kang, Keun-Wook Lee, Sun Min Lim, Jin-Soo Kim, Hyun Woo Lee, Min Kyoung Kim, In Gyu Hwang, Sangwoo Kim, Byung-Ho Nam, Hye Ryun Kim

**Affiliations:** 1Department of Medical Oncology, International St Mary’s Hospital, Catholic Kwandong University College of Medicine, Incheon 22711, Korea; hchang@ish.ac.kr; 2Division of Hematology and Medical Oncology, Department of Internal Medicine, Kangbuk Samsung Hospital, Sungkyunkwan University School of Medicine, Seoul 03181, Korea; gosciny@gmail.com; 3Division of Oncology, Department of Internal Medicine, Eunpyeong St. Mary’s Hospital, College of Medicine, The Catholic University of Korea, Seoul 03312, Korea; koyoonho@catholic.ac.kr; 4Division of Oncology, Department of Internal Medicine, Incheon St. Mary’s Hospital, College of Medicine, The Catholic University of Korea, Incheon 21431, Korea; wkdgh84@hanmail.net; 5Division of Hematology and Medical Oncology, Department of Internal Medicine, Konyang University Hospital, Daejeon 35365, Korea; jabuss@naver.com; 6Division of Hematology/Oncology, Department of Internal Medicine, Keimyung University Dongsan Hospital, Daegu 51448, Korea; kupark@dsmc.or.kr; 7Division of Oncology/Hematology, Department of Internal Medicine, Korea University College of Medicine, Seoul 02841, Korea; kkangju11@naver.com; 8Department of Internal Medicine, Seoul National University Bundang Hospital, Seoul National University College of Medicine, Seongnam 13620, Korea; hmodoctor@snubh.org; 9Division of Medical Oncology, Department of Internal Medicine, Yonsei University College of Medicine, Yonsei Cancer Center, Seoul 03722, Korea; limlove2008@yuhs.ac; 10Department of Internal Medicine, Seoul National University Boramae Medical Center, Seoul 07061, Korea; gistmd@gmail.com; 11Department of Hematology-Oncology, Ajou University School of Medicine, Suwon 16499, Korea; hwlee71@gmail.com; 12Division of Hematology-Oncology, Department of Internal Medicine, Yeungnam University College of Medicine, Daegu 42415, Korea; kmink21c@hanmail.net; 13Division of Hematology/Oncology, Department of Internal Medicine, Chung-Ang University Hospital, Chung-Ang University College of Medicine, Seoul 06973, Korea; oncology@cau.ac.kr; 14Department of Biomedical Systems Informatics and Brain Korea 21 PLUS Project for Medical Science, Yonsei University College of Medicine, Seoul 03722, Korea; SWKIM@yuhs.ac; 15HERINGS, The Institute of Advanced Clinical & Biomedical Research, Seoul 08378, Korea; byunghonam@heringsglobal.com

**Keywords:** *CD2000R1*, mRNA expression, prognosis, immune cell, head and neck cancer

## Abstract

Immune system dysfunction is associated with head and neck squamous cell carcinoma (HNSCC) development and progression and immune checkpoint inhibitors have demonstrated substantial survival benefits in platinum-refractory HNSCC; therefore, we examined the prognostic value of immune-related gene (IRG) expression in HNSCC. We analyzed the expression of 82 IRGs in 71 patients with HNSCC enrolled in a feasibility study for a prospective HNSCC biomarker-driven umbrella trial (Korean Cancer Study Group TRIUMPH study, NCT03292250). *CD200R1* was identified as an independent prognostic factor and validated in GEO and TCGA database. *CD2000R1* mRNA expression was found to be an independent favorable prognostic factor in patients with HNSCC. Moreover, *CD200R1* was found to affect genes and pathways associated with the immune response, while seven differentially expressed genes (*CD8A, DOK2, CX3CR1, TYROBP, CXCL9, CD300LF, IFNG*) were associated with *CD200R1* expression. Samples with higher *CD200R1* expression displayed higher tumor-infiltrating immune cell counts both in silico and in histological analysis. These findings will help in the development of more accurate prognostic tools and suggest CD200R1 modulation as a HNSCC immunotherapy.

## 1. Introduction

Head and neck squamous cell carcinoma (HNSCC) is one of the most common cancers worldwide, with more than 350,000 related deaths per year [1,2]. Despite the presence of established combined and multidisciplinary therapies against HNSCC, treatment outcomes remain poor and have improved little in recent decades [3]. Major prognostic factors associated with HNSCC, include invasive depth, cervical lymph node metastasis, and distant metastasis; however, the identification of clinical prognostic factors remains challenging despite increasing knowledge of cancer genomics [4]. Thus, new prognostic biomarkers for HNSCC are urgently needed to enable clinicians to select appropriate treatments.

The immune system is known to be significantly involved in HNSCC progression [5,6,7], and anti-programmed cell death protein 1 (PD-1) immune checkpoint inhibitors have been approved to treat recurrent and/or metastatic HNSCCs that are refractory to platinum therapy [8,9]. Several immune-biomarkers have been highlighted as useful prognostic indicators for HNSCC [10,11,12,13]; for instance, circulating CD8^+^ T cell levels prior to treatment have been found to be associated with improved survival in patients with advanced oropharyngeal cancer [10], while high tumor-infiltrating lymphocyte levels are related with increased overall survival (OS) [11], and increased CD68^+^ macrophage counts in oral squamous cell carcinoma are known to be significantly correlated with mortality [12]. However, the molecular characteristics of tumor-immune interactions remain largely unclear, and few studies have evaluated the prognostic value of immune-related genes (IRGs) in a large HNSCC database.

In this study, we explored the prognostic value of IRGs in patients with HNSCC from a retrospective Korean cohort, and validated their prognostic significance using data from the Gene Expression Omnibus (GEO) and The Cancer Genome Atlas (TCGA) databases. In addition, we investigated the potential mechanism of action of the IRGs, and quantified immune cell infiltration which was stratified based on IRG expression.

## 2. Results

### 2.1. Patient Characteristics

The clinicopathological characteristics and *CD200R1* mRNA expression status of the 71 patient tumors included in this study are shown in Table 1. The cohort included 58 (82%) men and 13 (18%) women between the ages of 28 and 81, and almost 29% of the patients had never smoked. Most of the tumors were human papillomavirus (HPV)-negative and more than 66% had reached advanced clinical stages (III or IV). The most common sites of tumor origin were the oral cavity (37%) and the oropharynx (24%).

### 2.2. IRG Expression and Survival of Patients with HNSCC

To determine the relationship between gene expression and prognosis, IRG expression was analyzed using a Nanostring assay and log-rank tests were conducted to evaluate the prognostic effect of IRG mRNA expression on OS (Appendix A). Among the 82 IRGs, 47 were significantly correlated with OS (*p* < 0.05) and underwent regularized Cox regression which identified 12 IRGs as being significantly associated with OS (Coxnet beta ≠ 0). To confirm the prognostic reproducibility and stability of the selected genes, we conducted external validation using TCGA and GEO (GSE65858) HNSCC data. *CD200R1* was identified as a significant OS-related factor in the Korean cohort, TCGA, and GEO data (Figure 1).

Next, a Cox proportional-hazard model was used to verify the independent prognostic value of *CD200R1* mRNA levels and well-known clinical factors. *CD200R1* mRNA expression was found to be an independent predictor of OS in the Korean cohort and TCGA and GEO data after adjusting for clinical features (Table 2). High *CD200R1* expression was correlated with longer median survival than low *CD200R1* expression in the Korean cohort [median survival 75.5 vs. 23.4 months, hazard ratio (HR): 0.18, (95% confidence interval (CI): 0.05–0.54; *p* < 0.01), TCGA data (median survival: 60.4 vs. 47.0 months, HR: 0.72, 95% CI: 0.53–0.98; *p* < 0.05), and GEO data (median survival: 64.6 vs. 42.3 months, HR: 0.52, 95% CI: 0.28–0.94; *p* < 0.05; Figure 1). These findings demonstrated that *CD200R1* expression was an independent prognostic IRG in all three HNSCC cohorts.

### 2.3. Performance of CD200R1 mRNA Expression as a Biomarker

To assess the prognostic performance of *CD200R1*, we compared its ability to predict OS with that of other clinical factors using Uno’s C-index and AUC values (Table 3). Compared to the other clinical factors, *CD200R1* mRNA expression yielded a higher C-index value for five-year survival in the Korean cohort alongside a higher AUC value, but exhibited similar performance for the GEO and TCGA data. These means that the performance of *CD200R1* was compatible with known clinical prognostic factors.

### 2.4. Differentially Expressed Genes (DEGs) and Gene set Enrichment Analysis

To explore the role of *CD200R1* in HNSCC, we determined *CD200R1* mRNA expression in normal and tumor tissues in TCGA datasets from the University of California, Santa Cruz Xena platform [14]. Significantly lower *CD200R1* expression (Welch’s *t*-test, *p* < 0.001) was observed in the tumor tissue than that in normal tissue (Figure 2).

To further elucidate the biological function of *CD200R1,* we performed differential expression (DE) analysis between tumors with high and low *CD200R1* expression from the GEO and TCGA datasets, respectively. A total of 130 genes were identified in the GEO data with a fold change > 1.5 and adjusted *p*-values of <0.05 using the Benjamini‒Hochberg method (Figure 3a).

To better understand the different underlying biological processes associated with these genes, we performed GO enrichment analysis on the DEGs that met our differential expression criteria in tumors with high and low *CD200R1* expression. We identified GO biological process terms, with a Benjamini-Hochberg corrected *p*-value of <0.05, which were summarized using a REVIGO tree map (Figure 4a and Appendix A). The enrichment of biological processes was mainly associated with the immune response and the regulation of leucocyte apoptosis.

In the TCGA data, 800 genes with a fold change > 2.0 and an FDR of <0.01 (Figure 3b) were subjected to GO enrichment analysis. A REVIGO map of these DEGs revealed that the immune response was an important enriched biological process (Figure 4b and Appendix A), while REVIGO analysis of 67 common DEGs between the GEO and TCGA data showed that the immune response and negative regulation of leukocyte apoptosis were essential biological processes (Figure 4c and Appendix A).

### 2.5. Interaction between CD200R1 and IRGs

We generated a set of genes known to interact with *CD200R1* by searching the STRING database and conducting a literature review [15,16,17]. Seven DEGs (*CD8A, DOK2, CX3CR1, TYROBP, CXCL9, CD300LF, IFNG*) in the TCGA data and one DEG (*CD8A*) in GEO were likely to be functionally related to *CD200R1* (Figure 3). Of these, *CD8A* was significantly correlated with *CD200R1* expression (Pearson correlation coefficient (*r*) = 0.20, *p* < 0.1 for Korean cohort; *r* = 0.47, *p* < 0.001 for GEO data; and *r* = 0.33, *p* < 0.001 for TCGA data). These suggest that *CD200R1* had interactions with other IRGs including *CD8A.*

### 2.6. Correlation of CD200R1 with Immune Cell Signatures

CD200R1 is strongly expressed in macrophages, neutrophils, and other leucocytes such as monocytes and mast cells; therefore, we deconvoluted gene expression in the GEO and TCGA data to estimate microenvironmental composition by ConsensusTME analysis. Significant enrichment in CD8^+^ and CD4^+^ T cells was observed in samples with high *CD200R1* mRNA expression compared to those with low *CD200R1* expression in both the GEO and TCGA datasets (Figure 5). Furthermore, M1 and M2 macrophage scores were significantly higher in samples with high *CD200R1* mRNA expression and higher immune scores were observed in tumors with high *CD200R1* expression. These indicated that tumors with high *CD200R1* expression have a ‘immune-rich’ microenvironment.

### 2.7. Association between CD200R1 and Lymphocyte Infiltration

To confirm the association between *CD200R1* expression and lymphocyte infiltration in tumor tissue, we examined TCGA patient cohorts with matched histology-based lymphocytes infiltration scores and gene expression data. A clear positive association was found between *CD200R1* expression and tissue-based quantitation of lymphocyte infiltrates (*r* = 0.16; *p* = 0.003) (Figure 6A), and significantly higher lymphocyte infiltration was detected in samples with high *CD200R1* expression than in those with low *CD200R1* expression (Welch *t*-test, *p* < 1 × 10e^−3^) (Figure 6B). These demonstrated that HNSCC with high *CD200R1* expression were characterized by a microenvironment rich in lymphocytes. 

## 3. Discussion

In this study, we aimed to identify IRG markers to predict the prognosis of patients with HNSCC. *CD200R1* was found to be an immune prognostic biomarker in the Korean HNSCC cohort and its predictive significance was systematically validated using the large GEO and TCGA databases. In particular, we demonstrated that patients with HNSCC and high tumor tissue *CD200R1* mRNA expression displayed significantly better OS than those with low *CD200R1* expression. Multivariate analysis with clinical factors further confirmed that *CD200R1* expression was an independent prognostic indicator in all three HNSCC cohorts. To explore the mechanisms of action of *CD200R1* in HNSCC, we systematically investigated the GEO and TCGA datasets and found that *CD200R1* was positively associated with an enhanced immune response. In silico and histological analyses revealed that samples with high *CD200R1* expression contained more tumor-infiltrating immune cells, while cyto-chemokine networks (*CX3CR1, CXCL9, IFNG)* and immuno-receptor and adaptor proteins (*DOK2, TYROBP, CD300LF*) were likely to contribute to antitumor cell immunity in tumors with high *CD200R1* expression.

Cox regression analysis of *CD200R1* expression could improve our understanding of the biological role of CD200R1 in cancer [18,19] as CD200R1 signaling is associated with tumor growth and metastasis [17,20]. CD200R1 deficiency was found to alter the tumor microenvironment (TME), leading to accelerated tumor growth and reduced T cell infiltration and/or effector function, while *CD200R1* expression was significantly lower in HNSCC tumors than that in normal tissues. These findings are consistent with a previous study on patients with gastric cancer, which found that the frequencies of CD3^+^, CD3^+^/CD4^+^, and CD3^+^/CD8^+^ T cells expressing CD200R1 were lower in the peripheral blood of patients with gastric cancer than that in the healthy control patients [21]. 

DAVID and REVIGO analyses revealed that DEGs were enriched for the immune response and the negative regulation of leucocyte apoptosis. A previous study using B16 melanoma and J558 plasmacytoma mouse syngeneic models found that *CD200* expression could inhibit tumor growth and metastasis by shaping the TME [17,20], while another indicated that CD200R1-deficient mice display accelerated CD200^+^ B16 melanoma growth [16]. Indeed, the livers of these CD200R1-deficient mice exhibited increased metastatic CD200^+^ tumor growth which contained higher numbers of CD11b^+^Ly6C^+^ myeloid cells, displayed VEGF and HIF1a gene expression with increased angiogenesis, and showed significantly reduced CD4^+^ and CD8^+^ T cell infiltration [16]. Together with our results, these studies linking CD200R1 and immune cell infiltration suggest a positive association between *CD200R1* expression and the immune response. Immune cell mRNA deconvolution using ConsensusTME showed that tumors with high *CD200R1* expression contained more tumor-infiltrating cells such as macrophages and T cells, consistent with the significantly higher histology-based lymphocyte infiltration score observed in samples with high *CD200R1* expression. These findings are particularly noteworthy as they were derived from different data types (i.e., mRNA vs. microscopy-based images).

Several potential mechanisms have been associated with CD200R1 and immune responses [17,20,22]. For instance, 4THM breast cancer cell growth and invasion were increased in CR200R1KO mice and decreased in mice over-expressing CD200, with a lack of CD200R1 expression related to decreased CD8^+^ and CD3^+^CD25^+^ T cells [23]. In contrast, CD200 overexpression in CD200 transgenic mice was found to increase tumor-induced IFN-γ and decrease inflammatory cytokine, TNF-α, and IL-6 expression. Moreover, CD200fc, which mimics the effect of CD200, significantly decreased tumor growth and metastasis in a mouse model of breast cancer, increased IFN-γ and decreased IL-6 expression, and time-dependently altered IL-10 and IL-17 levels [22]. In this study, we analyzed genes that are biologically related to *CD200R1* using the STRING database and by conducting a literature review. *CD8A, DOK2, CX3CR1, TYROBP, CXCL9, CD300LF*, and *IFNG* were identified as significant DEGs between tumor samples with high and low *CD200R1* expression.

CX3CR1 expression is observed in monocytes, macrophages, dendritic cells, T cells, and natural killer cells, while CX3CR1 is known to bind to CXCL6 and CXCL8 [24,25]. In addition, CX3CR1 levels have been shown to correlate with the degree of effector CD8^+^ T cell differentiation and CX3CR1^+^ cells are the predominant memory T cells in peripheral tissues [26]. CXCL9 is an IFN-γ-inducible chemokine that has been associated with the activation of T helper type 1 immunity in the TME and favorable responses to chemotherapy and immunotherapy in melanoma [27]. Although IFN-γ signaling has diverse biological functions, it is primarily related to host defense and immune regulation, including anti-viral and anti-bacterial defense, anti-tumor functions, cell cycle, apoptosis, inflammation, and innate and acquired immunity [28]. *DOK2* encodes docking protein 2 which is preferentially expressed in hematopoietic cells and is involved in the negative regulation of signaling pathways downstream of various immunoreceptors, such as B cell, T cell, Fc, and Toll-like receptors [29]. Moreover, DOK2 can bind to the intracellular domain of CD200R1 and recruit the Ras-GTPase activator protein which is required for inhibition of myeloid cell activation by CD200R1 [30]. TYRO is a protein tyrosine kinase-binding protein encoded by *TYROBP* that is an immunoreceptor tyrosine-based activation motif (ITAM)-bearing transmembrane adaptor protein, which is expressed in NK cells, granulocytes, monocytes, macrophages, dendritic cells, and mast cells [31]. TYRO signals via the ITAM within its cytoplasmic domain and induces cellular activation characterized by cytokine production, proliferation, and survival in macrophages and antigen presentation, maturation, and survival in dendritic cells [32]. Finally, CMRF35-like molecule 1, encoded by *CD300LF,* acts as an inhibitory receptor in myeloid cells and mast cells [33], with anti-human CMRF35-like molecule 1 antibodies displaying complement- and antibody-dependent cell cytotoxicity against CMRF35-like molecule 1-expressing AML-derived cell lines and freshly isolated blasts from AML patients in vitro [34]. 

Collectively the associations between CD200R1 and these diverse immune-related proteins suggest that CD200R1 is involved in a variety of immune mechanisms and may explain why HNSCC with high *CD200R1* expression is related to an ‘immune-rich’ microenvironment with high immune cell estimates. Thus, our findings suggest that modulation of the CD200R1 pathway could be used as an immunotherapeutic strategy in patients with HNSCC [22,35,36].

## 4. Materials and Methods

### 4.1. Patients and Data Collection

We analyzed 71 patients with HNSCC enrolled in a feasibility study for a prospective HNSCC biomarker-driven umbrella trial (Korean Cancer Study Group TRIUMPH study, NCT03292250). The data collection methods have been described previously [37]. Briefly, pretreatment tumor tissues and matched normal DNA were obtained from retrospectively recruited patients with HNSCC between 2016 and 2017 through the approval of the Institutional Review Boards of 19 institutions in the Republic of Korea. Clinicopathological data were collected from the electrical medical records of the patients in accordance with an IRB-approved protocol (NCT03292250); these included age, sex, the anatomic site of the tumor, tobacco and alcohol use, clinical stage based on the American Joint Committee on Cancer (AJCC) 7th edition, treatment history, and survival data. All patients provided written informed consent for the genomic testing conducted for this study.

### 4.2. Nanostring Assay

Briefly, total tumor RNA was isolated using an RNeasy kit (Qiagen, Valencia, CA, USA) [37]. NanoString nCounter assays were performed according to the manufacturer’s standard protocol and raw data were normalized using nSolver Analysis Software version 4.0 (NanoString Technologies, Seattle, WA, USA) and log2-transformed for further analysis. Twenty-two samples with normalization flags were removed prior to final analysis. The remaining 71 samples were finally included in the present analysis. The complete list of 82 IRGs is shown in Appendix A.

### 4.3. TCGA and GEO Data Analysis

mRNA expression and clinical data were downloaded from TCGA HNSCC database in January 2019 [38,39] and analyzed using R statistical software with the cgdsr package. The comparison of *CD200R1* mRNA expression between tumor and normal tissues were analyzed by using the UCSC Xena browser (http://xena.ucsc.edu/) [14]. The GSE65858 dataset analyzed using, Illumina HumanHT-12 V4.0 Expression Beadchips (Illumina, San Diego, CA, USA) included 270 HNSCC tumor samples with clinical and prognostic variables and was downloaded from the GEO database (www.ncbi.nlm.nih.gov/geo/) using the *GEOquery* R package in January 2019 [40,41].

### 4.4. Statistical Methods

Statistical evaluation is summarized in Appendix A. In the training phase with the Korean cohort, the Kaplan‒Meier method with log-rank test was used for OS analysis to elucidate IRG expression. OS was measured from the date of diagnosis until the date of death from any cause, excluding patients who were alive at their last follow-up. Patients were divided into low- and high-expression groups using optimal cut-off points for IRG expression determined using R software with the MaxStat package, which computes the maximal log-rank statistic to identify the cutoff that provides the best separation (i.e., where standardized statistics are maximal) [42]. Kaplan‒Meier analysis was performed to estimate the survival curves of the low- and high-expression groups of each IRG and the log-rank test was used to compare the two curves. To narrow down the candidate genes, IRGs with *p*-values of < 0.05 from the log-rank test were selected for regularized Cox regression, which is a well-established method for selecting the most predictive markers for time-to-event analysis [43]. 

During the validation phase, the IRGs selected by regularized Cox regression were analyzed with GEO (GSE65858) and TCGA HNSCC data using the Kaplan‒Meier method and log-rank test. IRG cut-off points were calculated with maximally selected rank statistics using MaxStat, and the significantly validated IRGs were evaluated by multivariate Cox regression analysis to compare the effects of IRG expression and well-known clinical variables on prognosis.

Two methods were used to evaluate biomarker performance: Uno’s C-index in time-dependent area under the curve (AUC) analysis; AUC values in receiver operating characteristic curves at the five-year mark [44]. These values were obtained using the R packages survC1 and epiDisplay.

### 4.5. DEG Screening and Analysis

To identify DEGs associated with the selected IRGs, patients were divided into groups based on *CD200R1* mRNA expression (high and low). DEGs in the TCGA data were determined using the R package TCGAbiolinks with the R/Bioconductor package *edgeR* [45,46]. DEGs were selected in the GEO data using the R/Bioconductor limma package [47,48]. Adjusted *p*-values for multiple testing were calculated using Benjamini-Hochberg correction.

The GO term enrichment analysis of subnetwork genes was carried out using DAVID Bioinformatics Resources 6.818 [49] along with the results of the GOTERM_BP_DIRECT approach with a Benjamini adjusted *p*-value of 0.05 or less. GO terms were summarized using REVIGO (available at http://revigo.irb.hr), excluding redundant GO terms and visualizing the remaining terms in a tree map. The *p*-values of the enriched GO terms were used to determine the size of tree maps.

PPI network analysis was conducted using the STRING database (http://string-db.org/) to identify proteins related to those encoded by the selected IRGs [15].

### 4.6. Immune Cell Analysis with In Silico Deconvolution and Microscopic Examination

The ConsensusTME method integrates cell type-specific gene markers from independent TME cell estimation methods, and uses single sample Gene Set Enrichment Analysis (ssGSEA) to compute relative TME cell type- and tumor-specific enrichment scores from bulk expression data [50]. This method has previously been thoroughly benchmarked in a pan-cancer setting. Normalized enrichment scores (NSE) for each cell type in each sample in the GEO (GSE65858) and TCGA HNSCC datasets were calculated with gene sets specific for human head and neck carcinoma. Histology-based lymphocyte infiltration scores were obtained as a percentage of the tumor from TCGA via the Genomic Data Commons (https://gdc.cancer.gov/).

## 5. Conclusions

This study had some limitations. Firstly, the patient population was heterogeneous due to the retrospective nature of this study. Secondly, training phase analysis was performed using a Nanostring panel encompassing selected key genes expressed in the TME. As the TME contains various immune system components, HNSCC prognosis cannot be absolutely defined by a single factor, resulting in the relatively low performance (C-index) of *CD200R1* and other clinical factors in our study. Nevertheless, *CD200R1* expression was confirmed to be a robust and independent prognostic factor by multivariate analysis, alongside several other clinical characteristics in the Korean, GEO, and TCGA cohorts. Further studies that analyze the prospectively-collected tissue samples with large patient numbers are required to validate our results. In conclusion, our findings suggest that *CD200R1* mRNA expression is an independent favorable prognostic factor in HNSCC. This information increases our understanding of the molecular biology of HNSCC and may allow the development of tools to more accurately predict patient prognosis. In addition, our study indicates that CD200R1 may be a potential immunotherapeutic target for HNSCC.

## Figures and Tables

**Figure 1 cancers-12-01777-f001:**
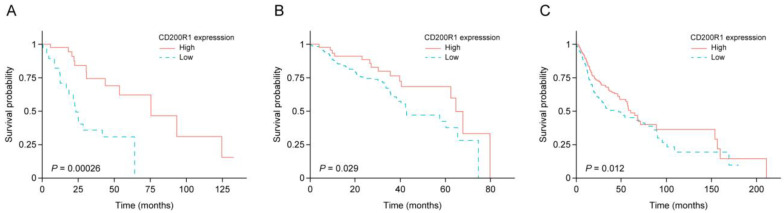
Kaplan–Meier survival curve for changes in OS according to *CD200R1* mRNA expression in the Korean cohort (**A**), the GEO (GSE65858) (**B**), and TCGA HNSCC (**C**) data. Patients with high expression of *CD200R1* (yellow solid line) showed longer OS compared with those with low expression (blue dotted line). The *p*-values were calculated using the log-rank test.

**Figure 2 cancers-12-01777-f002:**
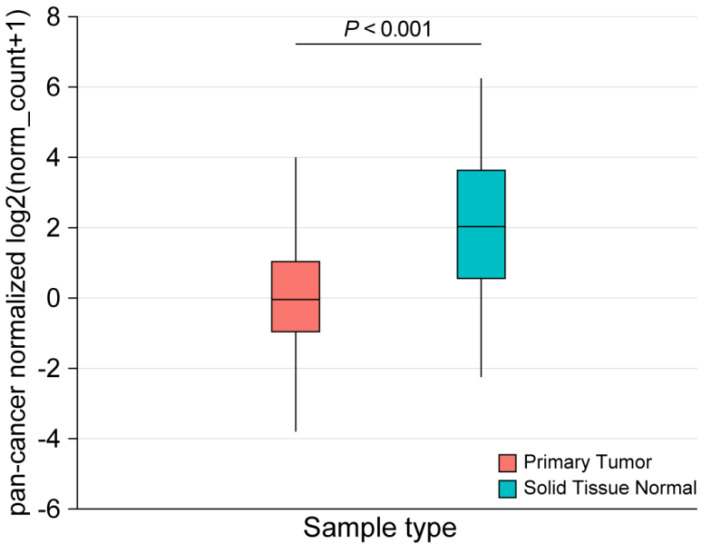
*CD200R1* mRNA expression in tumors (blue box) and normal tissues (purple box) in TCGA HNSCC (*n* = 602) using the Xena platform (Welch’s *t*-test, *p* < 0.001).

**Figure 3 cancers-12-01777-f003:**
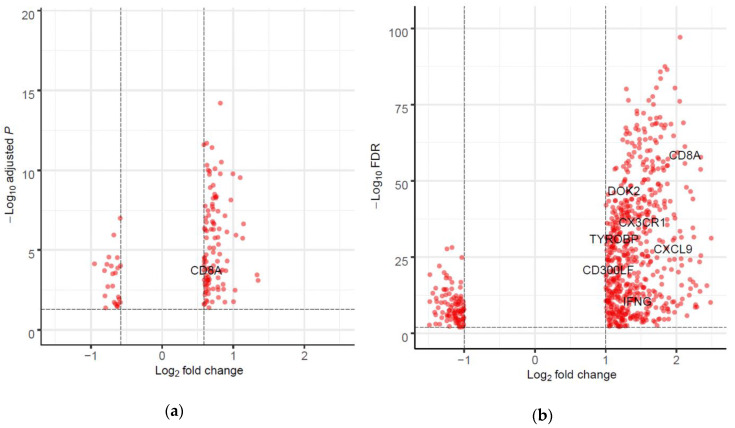
Volcano plots of all differentially expressed genes (DEGs) between tumors with high and low *CD200R1* expression. (**a**) DEGs in GEO dataset. Red points represent genes with a |fold change| of >1.5 and adjusted *p*-values of <0.05. Dotted horizontal line indicates an adjusted *p*-value of <0.05. Dotted vertical line indicates a |fold change| of >1.5. (**b**) DEGs in TCGA dataset. Red points represent genes with a |fold change| of >2.0 and a false discovery rate (FDR) of <0.01. Dotted horizontal line indicates an FDR < 0.01. Dotted vertical line indicates a |fold change| of >2.0.

**Figure 4 cancers-12-01777-f004:**
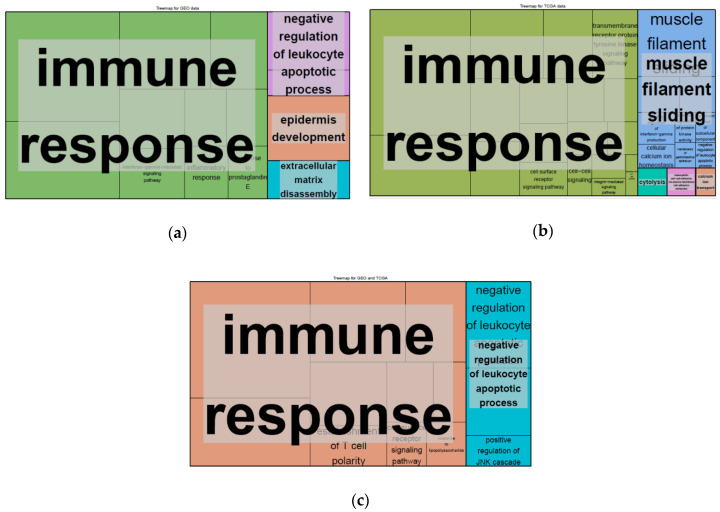
REVIGO tree maps of gene ontology terms for enriched biological processes in differentially expressed genes (DEGs) associated with *CD200R1* expression in the GEO (GSE65858) (**a**) and TCGA HNSCC (**b**) data. (**c**) Common DEGs associated with *CD200R1* expression in the GEO (GSE65858) and TCGA HNSCC data. Box color indicates the functional category and box size is proportional to its corrected *p*-value.

**Figure 5 cancers-12-01777-f005:**
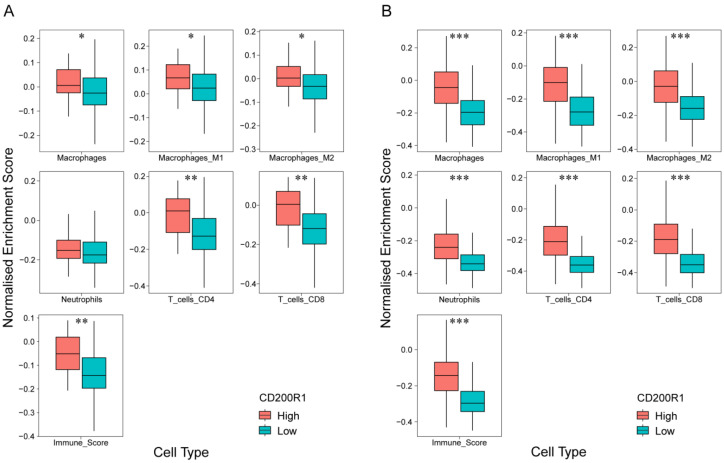
Boxplots of immune cell gene scores (normalized enrichment scores) stratified based on *CD200R1* mRNA expression in the GEO (**A**) and TCGA (**B**) datasets using ConsensusTME analysis. (**A**) * *p* < 0.001, ** *p* < 1 × 10e^−5^; (**B**) *** *p* < 2 × 10e^−16^.

**Figure 6 cancers-12-01777-f006:**
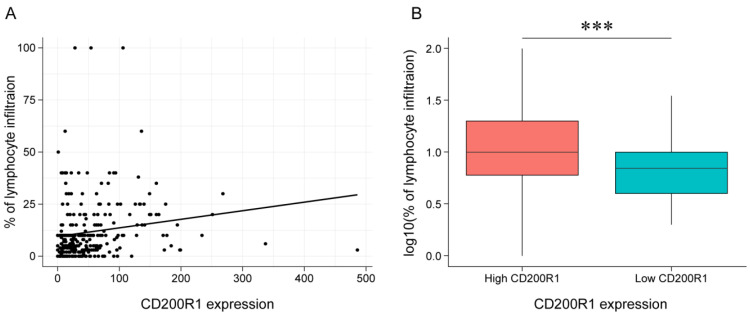
*CD200R1* mRNA expression and tumor lymphocyte infiltration in the TCGA HNSCC. A plot for correlation analysis (**A**). Pearson correlation calculation was performed for linear correlation. A plot for mean comparison stratified based on *CD200R1* mRNA expression (**B**) *** *p* < 1 × 10e^−3^.

**Table 1 cancers-12-01777-t001:** **C**linical and pathological characteristics and *CD200R1* mRNA expression in the Korean HNSCC cohort.

Patient Characteristics		No.	%	High *CD200R1* Expression No./Subgroup (%)
**Total**		71	100	43/71 (60.5)
Sex	Male	58	81.6	35/58 (60.3)
	Female	13	18.4	8/13 (61.5)
Age	<58 years	31	43.6	19/31 (61.2)
	≥58 years	40	56.3	24/40 (60.0)
Smoking	Never	20	28.1	10/20 (50.0)
	Yes	51	71.8	33/51 (64.7)
Tumor location	Oral cavity	26	36.6	15/26 (57.6)
	Oropharynx	17	23.9	11/17 (64.7)
	Hypopharynx	10	14	7/10 (70.0)
	Larynx	12	16.9	8/12 (66.7)
	Other	6	8.5	2/6 (33.3)
Stage	I	11	15.5	8/11 (72.7)
	II	13	18.3	8/13 (61.5)
	III	17	23.9	8/17 (47.0)
	IV	30	42.3	19/30 (63.3)
Human papilloma virus	Negative	47	66.2	26/47 (55.3)
	Positive	12	16.9	10/12 (83.3)
	Unknown	12	16.9	7/12 (58.3)

**Table 2 cancers-12-01777-t002:** Multivariate analysis of the overall survival in the Korean cohort and the GEO (GSE65858) and TCGA HNSCC data.

Factors	Korean Cohort		GEO (GSE65858)		TCGA HNSCC	
	*p* Value	Hazard Ratio (95% CI)	*p* Value	Hazard Ratio (95% CI)	*p* Value	Hazard Ratio (95% CI)
*CD200R1* (high vs. low)	<0.01 *	0.19 (0.06–0.58)	<0.05 *	0.54 (0.29–0.99)	<0.05 *	0.72 (0.53–0.98)
Smoking (Yes vs. never)	>0.1	1.67 (0.37–7.48)	>0.1	0.97 (0.55–1.70)	-	-
T stage (T3 + T4 vs. T1 + T2)	>0.1	1.12 (0.95–1.31)	0.001 *	2.81 (1.49–5.30)	<0.001 *	1.74 (1.25–2.43)
N stage (N+ vs. N0)	>0.1	3.06 (0.53–17.7)	>0.1	1.14 (0.63–2.03)	<0.0001 *	2.00 (1.43–2.78)
M stage (M1 vs. M0)	>0.1	6.27 (0.36–115.9)	>0.1	1.91 (0.76–4.75)	>0.1	1.01 (0.98–1.05)
Stage (III, IV vs. I, II)	>0.1	0.52 (0.09–3.01)	>0.1	0.93 (0.33–2.59)	-	-
Tumor grade (2, 3 vs. 0, 1)	>0.1	0.94 (0.81–1.09)	-	-	-	-
Sex (F vs. M)	>0.1	2.89 (0.84–9.90)	>0.1	1.00 (0.58–1.73)	>0.1	0.81 (0.58–1.12)
Age	>0.1	1.00 (0.95–1.06)	>0.1	1.18 (0.76–1.84)	>0.1	1.33 (0.97–1.80)

******p* value < 0.05. CI: confidence interval.

**Table 3 cancers-12-01777-t003:** C-index and AUC values for the prediction of five-year survival by *CD200R1* and clinical factors in the Korean cohort and the GEO (GSE65858) and TCGA HNSCC data.

Factors	Korean Cohort			GEO Data			TCGA HNSCC Data		
	C-index	95% CI	AUC	C-index	95% CI	AUC	C-index	95% CI	AUC
*CD200R1*	0.59	0.41–0.77	0.63	0.58	0.53–0.63	0.58	0.53	0.49–0.57	0.54
T stage	0.55	0.42–0.69	0.50	0.58	0.51–0.65	0.58	0.55	0.52–0.59	0.58
N stage	0.55	0.41–0.70	0.59	0.53	0.45–0.60	0.55	0.56	0.52–0.59	0.58
Age	0.52	0.39–0.66	0.50	0.64	0.43–0.81	0.62	0.52	0.47–0.56	0.53
Sex	0.54	0.44–0.64	0.59	0.58	0.42–0.74	0.58	0.59	0.49–0.57	0.54

CI: confidence interval.

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
