# Peer review of "Prognostic Value of CD200R1 mRNA Expression in Head and Neck Squamous Cell Carcinoma"

_cancers, 2020, doi:10.3390/cancers12071777_

Round 1

Reviewer 1 Report

The authors prepared 3 kinds of database. One is a data from their own patients with HNSCC between 2016 and 2017. They accumulated clinical and histopathological characteristics and obtained tumor tissues and normal DNA from 71 patients (strangely the patients number in Abstract part was written as 93). The others were data from Gene Expression Omnibus (GEO) and The Cancer Genome Atlas (TCGA), which were introduced from the accessible database.

Initially, the authors examined their own specimens about 82 immune related genes (IRGs) and pick up 47 IRG affecting overall survival of their patients. Regularized Cox regression analysis was made to focus on and 12 IRGs remained. The authors called this process “training phase”.

Secondary, the authors use the other databases to validate 12 IRGs they focused on the training phase. CD200R1 gene is only factor affecting overall survival in every database.

In the next part, the authors compared the prognostic value of CD200R1 with other clinical characteristics known to influence on the patient’s survival. CD200R1 appeared to be only factor showing better prognosis in every database among smoking condition, T stage, N stage, M stage, clinical stage, age and sex. Uno’s C-index and AUC values demonstrated that the performance of CD200R1 was compatible with known clinical prognostic factors.

The later part of the results was obtained only from the data of GEO and TCGA.

The authors revealed CD200R1 was expressed lower in the tumor tissue than the corresponding normal tissue using TGCA date. CD200R1 gene was proved to be associated with many other genes related to immune response using GEO and TCGA data. The consensus tumor microenvironment (TME) analysis showed that more immune related cells, such as CD8+ and CD4+ T cells, macrophages, neutrophils accumulated in the tumor with high CD200R1 expression than those with low CD200R1 in GEO and TCGA data. More tumor lymphocyte infiltration (TIL) was seen in the tumor with high CD200R1 expression in TCGA data.

The results of the present study increase the readers understanding of the molecular biology and the tumor microenvironment of HNSCC. CD200R1 seems to be a promising factor predicting patients’ survival and may be a factor predicting the effectiveness of existing immunotherapy. Moreover this gene will be a potential immunotherapeutic target for HNSCC as the authors concluded.

Major points.

#1. The patients’ number which was described 71 in the body of the manuscript appeared 93 in the abstract part.

#2. The explanation of the result obtained from TCGA dataset was not seen in the chapter 2.4. Differentially expressed genes (DEGs) and gene set enrichment analysis. There should be some explanation because there is a figure showing this result (Figure 3(b)).

#3. The authors used TNM classification for the clinical characteristics of their own data and the existing databases. As we know, UICC/AJCC TNM classification has been changing overtime. Especially, the changes from 7Th edition to 8th edition were so drastic. The authors should confirm the edition of TNM classification they utilize was the same as the edition used in GEO and TCGA database. Moreover, the TNM edition the authors used should be described in the manuscript because it was updated 8th in 2017.

#4. There are 3 kinds of box plots in the manuscript. All of them have their own colors, shapes, proportions and font sizes.

It should be unified in one manuscript.

Minor points.

P2L62: , and metastasis;

This should be distant metastasis

P8L236: to increase tumor-induced INF-r

This may be IFN-r

Author Response

Point-by-point response to the recommendations of reviewer #1

We appreciate your peer review and valuable comments on our manuscript. We are very pleased with your positive responses.

Major points

Reviewer #1: The patients’ number which was described 71 in the body of the manuscript appeared 93 in the abstract part.

⇒ As you pointed out, the number of patients was corrected to 71 in the abstract. (line 46) Among 93 samples in the feasibility study for TRIUMPH trial, we removed twenty-two samples with normalization flags from nSolver Analysis software. Finally 71 samples were analyzed in this study. We described it in more detail in the method. (lines 289-290)

Reviewer #1: There should be some explanation because there is a figure showing this result.

⇒ We described the TCGA result in lines 151-156.

Reviewer #1: The authors should confirm the edition of TNM classification they utilize was the same edition used in GEO and TCGA database. Moreover, the TNM edition the authors used should be described in the manuscript.

⇒ As you pointed out, we described that the stage in our data was based on the AJCC 7th edition in the method in lines 281-283. (1) In addition, we confirmed that both GEO (GSE65858) and TCGA data were not based on AJCC 8th edition. (2, 3) We collected the originally reported stage in GEO and TCGA data. The patients were categorized into two groups for stage [high-stage (III, IV) disease vs. low-stage (I, II)] and analyzed for OS with other factors. (4) Given the introduction for high-risk HPV-associated oropharyngeal cancers and extranodal extension in the new edition, it could be suggested that further prospective studies should validate our results with the stage based on AJCC 8th edition. (5)

References

1) Sobin LH, et al. TNM Classification of Malignant Tumours, UICC, 7th edn. New York: Wiley-Liss, 2009.

2) Wichmann G, et al. The role of HPV RNA transcription, immune response‐related gene expression and disruptive TP53 mutations in diagnostic and prognostic profiling of head and neck cancer. Int. J. Cancer, 137: 2846-2857, 2015.

3) Lawrence, M. et al. Comprehensive genomic characterization of head and neck squamous cell carcinomas. Nature, 517:576–582, 2015.

4) Jianfang et al. An Integrated TCGA Pan-Cancer Clinical Data Resource to Drive High-Quality Survival Outcome Analytics. Cell, 173: 400–416.e11., 2018.

5) William Lydiatt, et al. Major Changes in Head and Neck Staging for 2018. American Society of Clinical Oncology Educational Book. 38:505-514, 2018.

Reviewer #1: It should be unified in one manuscript.

⇒ We revised the box plots and unified them.

Minor point

Reviewer #1: P2L62

⇒ It was corrected as distant metastasis. (line 63)

Reviewer #1: P8L236

⇒ We corrected it. (line 237)

Reviewer 2 Report

The manuscript by Chang et al., reports on the potential prognostic value of CD200R1 mRNA expression in head and neck squamous cell carcinoma (HNSCC). The molecular, bioinformatics and statistical analysis are all well designed and thorough and the results are interesting. 

However, there are some points of concern: 

  1. In the abstract, the number of patients with HNSCC is 93, whereas in the patient characteristics’ paragraph (line 85) the authors say that the patient tumors included in the study are 71. If we sum all numbers in table 1, the total is 71. If indeed the number of patients is 71 and not 93, then the authors should not use publication by Lim et al (14) for reference regarding patient's data (line 275), but instead prepare a new table including the data of patients used in this specific study. 
  2. In the same publication by Lim et al, the authors report that HPV-positive tumors were significantly associated with immune related genes (IRGs) when compared to HPV-negative tumors. Have the authors made an attempt to examine if the same conclusion applies in the present study regarding CD200R1 mRNA expression? 
  3. Sentences in lines 80-82 and 103-104 (figure 1 legend) should be rephrased. 
  4. Line 85 should be rephrased in order to include “high CD200R1 expression” to better describe the content of Table 1. 
  5. Please, provide figures with higher resolution. 
  6. DEG abbreviation should be spelled on the first time it appears. 
  7. In line 102, please delete “and” {(a) and the GEO...}. 
  8. In line 166, please change “implicated” to “suggests”. 

Author Response

Point-by-point response to the recommendations of reviewer #2

We appreciate your peer review and valuable comments on our manuscript. We are very pleased with your positive responses.

Reviewer #2: In the abstract, the number of patients with HNSCC is 93, whereas in the paragraph the patients in the study are 71.

⇒ As you pointed out, the number of patients was corrected to 71 in the abstract. (line 46) Among 93 samples in the feasibility study for TRIUMPH trial, we removed twenty-two samples with normalization flags from nSolver Analysis software. Finally 71 samples were analyzed in this study. We described it in more detail in the method. (lines 289-290) The method in patients and data collection was rephrased. (lines 274-276) We prepared a table including the data of patients which were analyzed in this specific study. (Table 1. with 71 patients for the present study)

Reviewer #2: if the same conclusion applies in the present study regarding CD200R1 mRNA expression?

⇒ As you mentioned, we evaluated the CD200R1 mRNA expression and HPV infection. The patients with HPV tended to have high CD200R1 expression, but only statistically significant in TCGA data.

Reviewer #2: Lines 80-82 and 103-104

⇒ As you pointed out, we rephrased the sentences. (lines 80-82, lines 103-105)

Reviewer #2: Lines 85

⇒ As you pointed out, we rephrased the sentences with CD200R1 expression. (lines 85-86)

Reviewer #2: figures with high resolution

⇒ We provided the figures with higher resolution.

Reviewer #2: DEG abbreviation

⇒ We corrected it in the abstract. (line 51)

Reviewer #2: delete ‘and’ in line 102

⇒ We deleted ‘and’ in line 102.

Reviewer #2: change ‘implicated’ to ‘suggest’

à We changed to ‘suggest’. (line 168)

Reviewer 3 Report

This study retrospectively investigated prognostic value of immune related genes in patients with HNSCC from a Korean cohort. They found that CD200R1 was identified as a significant favorable prognostic factor in the Korean cohort. This result was validated using data from the Gene Expression Omnibus (GEO) and The Cancer Genome Atlas (TCGA) databases. CD200R1 was found to affect genes and pathways associated with the immune response. Several differentially expressed genes were associated with CD200R1 expression. Overall, this study was well designed and data interpretation was fine.

Minor point:

  1. Please show us the methodology and whether mRNA or protein were analyzed in your experiment of CD200R1 expression in normal and tumor tissues in Figure 2.

Author Response

Point-by-point response to the recommendations of reviewer #3

We appreciate your peer review and valuable comments on our manuscript. We are very pleased with your positive responses.

Reviewer #3: The methodology and whether mRNA or protein was analyzed in the study.

⇒ We clearly described the methodology in the method. (lines 294-295) Also we explained that CD200R1 mRNA expression was analyzed between tumor and normal tissue. (lines 129, 133 and 294-295)